# Precision engineering of nano-assemblies in superfluid helium by the use of van der Waals forces
Gokhan Topcu[1], Aula M. A. Al Hindawi[1,2], Cheng Feng[1], Daniel Spence[1], Berlian Sitorus[1,3], Hanqing Liu[1], Andrew M. Ellis[1] & Shengfu Yang [1] ✉

The ability to precisely engineer nanostructures underpins a wide range of applications in areas such as electronics, optics, and biomedical sciences. Here we present a novel approach for the growth of nanoparticle assemblies that leverages the unique properties of superfluid helium. Unlike viscous solvents at or near room temperature, superfluid helium provides an unperturbed and cold environment in which weak van der Waals interactions between molecular templates and metal atoms become significant and can define the spatial arrangement of nanoparticles. To demonstrate this concept, diol and porphyrin-based molecules are employed as templates to grow gold nanoparticle assemblies in superfluid helium droplets. After soft-landing on a solid surface to remove the helium, transmission electron microscopy (TEM) imaging shows the growth of gold nanoparticles at specific binding sites within the molecular templates where the interaction between gold atoms and the molecular template is at its strongest.

Nanoparticles (NPs) have existing and potential technological applications in diverse areas such as sensors, catalysis, quantum devices and medical sciences. The ability to assemble NPs into an ordered structural arrangement can yield properties distinct from those exhibited by individual NPs. Self-assembly is a versatile approach for assembling colloidal particles where long-range van der Waals forces play a predominant role in driving the aggregation. However, a high level of control is required to harness these assemblies in many technological applications and facilitate efficient scale-up. In this regard, strategies based on wet-chemical synthesis, such as directed self-assembly and templated assembly of nanostructures, have been widely explored[1–4].

Templated synthesis of NP assemblies is a bottom-up approach that can guide the formation of specific nanoscale architectures. So far, a wide variety of templates have been explored, encompassing colloidal, polymeric, and molecular approaches[5]. Colloidal and polymeric strategies generally require relatively large templates, such as porous structures and micelles, to serve as scaffolds for creating NP assemblies. In contrast, molecular templates may offer nanoscale precision to assemble NPs without the involvement of additional procedures. For instance, supramolecular entities, such as thiol-modified DNA, have been exploited as templates to create NP assemblies via covalent bonding[5,6]. By capitalizing on the chemically active sites present in these molecules, in-situ synthesis can be achieved, facilitating a direct synthesis process.

Less explored is the use of non-covalent interactions with a template for creating NP assemblies. Harnessing these far weaker forces offers the potential to create more dynamic and tuneable structures due to their weaker and more reversible binding compared to covalent bonds, allowing particles to reorganize and thereby circumventing kinetic traps during the assembly process. Non-covalent interactions, such as hydrogen bonding, π–π stacking, hydrophobic interactions, host–guest binding, and van der Waals interactions, can drive the formation of ordered nanostructures[7]. For example, a handful of biological and organic templates, such as DNA[8,9], peptides[10,11], proteins[12,13], and polymers[14–16], have been utilized to facilitate the nucleation and growth of nanoparticle assemblies *via* non-covalent bonding. A typical procedure begins with the self-assembly of discrete molecules or subunits into larger supramolecular structures, which effectively function as templates. These templates are then exposed to metal ions that selectively associate with specific environments within the template. Metal nanostructures then grow when the metal ions are reduced by the addition of reducing agents[4].

Helium droplets (HeDs) can act as cryogenic nanoreactors[17,18]. The formation of small metal clusters in HeDs was first reported nearly three decades ago[19–22], but the use of HeDs as nanoreactors to produce sizeable metallic clusters, namely nanoparticles, emerged more recently[23–28]. In this regard, helium droplets possess a unique combination of properties, including (1) a very low steady-state temperature (0.37 K); (2) superfluidity,

[1]School of Chemistry, University of Leicester, Leicester LE1 7RH, UK. [2]Department of Chemistry, College of Education for Pure Science, University of Karbala, Karbala, Iraq. [3]Department of Chemistry, Tanjungpura University, Pontianak, Indonesia. ✉e-mail: sfy1@leicester.ac.uk

allowing unhindered motion of dopants and thus their aggregation into nano-clusters; and (3) the ability to form core–shell structures by sequential addition of materials to the droplets. HeDs also offer a means of soft-landing[29,30], allowing the helium to be removed and keeping any embedded nanostructures largely intact on a solid surface. Importantly, the superfluid helium offers rapid continuous cooling so that any vaporizable materials can potentially be incorporated into nanostructures, including elemental metals, semiconductors, liquid and, potentially, gaseous molecules, making HeDs a highly versatile medium for producing surfactant-free nanostructures[31–34].

Wet-chemical and high-temperature preparation of nanoparticle assemblies inevitably involves thermal effects, for example, collisions with other molecules, which affects the stability of the assemblies. As such, assembling nanoparticles using non-covalent forces necessitates relatively large interparticle distances to prevent particle agglomeration[35]; while for small templates that allow shorter interparticle distances, covalent interactions are needed to stabilize the assemblies[36]. In contrast, superfluid HeDs offer an isolated and unperturbed environment where the solvent is inert and non-viscous, and the temperature is exceptionally low. The suppression of thermal influence by the continuous and rapid cooling of superfluid helium then allows weak van der Waals interactions, whose effect would be negligible under the conditions used in typical wet-chemical or high-temperature synthesis, to become prominent. Since metal atoms are free to move within the superfluid, we assume that they will tend to bind to those sites on the molecular templates where the van der Waals force is the most strongly attractive. The addition of more metal atoms to the HeDs will then result in nanoparticles being grown at these sites. If this scenario is correct, the addition of metal atoms to a helium droplet containing a molecular template with more than one favoured binding site should lead to the formation of a well-defined nanoparticle assembly that is stable inside the droplet (see Fig. 1 for an illustration of this mechanism by using a diol molecule, with the two OH groups at the opposite ends of the molecule, as the exemplar template). Consequently, the use of superfluid HeDs may provide scope for the formation of new types of NP assemblies.

In this study, we explore this concept to assess its feasibility, flexibility and precision. We have employed two elongated chain diols, namely, 1,6-hexanediol and 1,8-octanediol, along with a porphyrin-based molecule, 5,10,15,20-tetra(4-pyridyl) porphyrin (H2TPyP), as the templates for construction of gold nanoparticle assemblies. Each diol molecule contains two O atoms, while the H2TPyP molecule comprises four pyridyl N atoms, all of which possess lone electron pairs. These atoms are expected to provide the strongest bonding to Au atoms because of the diffuse, polarizable outer valence $6s$ orbital of an Au atom, something confirmed by supporting calculations in this work.

## Results and discussion
### Diol-templated growth of nanoparticle assemblies
Figure 2 provides a visual representation of the process occurring in the helium droplet apparatus, wherein nanoparticle assemblies are formed through the sequential introduction of molecular templates and metal atoms (gold in this case) into the droplets. Subsequently, the HeDs collide with a solid surface, where the contents are deposited and can be imaged ex situ by using transmission electron microscopy (TEM).

We first consider the formation of gold NP assemblies by using diols as the templates. A low template doping rate is employed in these experiments to minimize the likelihood of having multiple diol molecules in one droplet. Under the conditions chosen, most of the HeDs (ca. 80–90%) within the droplet beam are statistically unlikely to contain any diol (referred to below as "empty droplets") and the rest predominately contain a single diol molecule. As such, the deposits on the lacey carbon TEM grids (~10 nm of thickness) will include gold nanoparticles formed in both empty droplets and those with a diol template, thereby ensuring identical experimental conditions and facilitating direct comparison. Another important consideration is the evaporative loss of helium atoms caused by the pickup of a diol molecule. Under the expansion conditions used for making the HeDs, namely a nozzle temperature of 8.5 K and a helium stagnation pressure of 15 bar, the HeDs

comprise $\sim 1.4 \times 10^6$ helium atoms, on average[37]. Adding a hexanediol or octanediol molecule will result in the loss of no more than 2000 helium atoms according to their thermal capacities[38]. Hence, the reduction in droplet size by adding a diol molecule is negligible. We, therefore, expect the same overall Au content (on average) for the empty HeDs and those containing a diol molecule. Finally, a relatively short deposition time, i.e., 10 s, has been employed to avoid excessive surface coverage by the NPs. This minimizes the probability of NPs landing randomly in close proximity to each other, making it easier to distinguish genuine NP pairs from the single NPs formed in empty droplets in TEM images.

TEM images of Au NPs, with and without the addition of 1,6-hex-anediol and 1,8-octanediol, are shown in Fig. 3. Without the diols, the distribution of NPs on the TEM grid is random (see Fig. 3a). When a molecular template is added the situation changes (see Figs. 3b, c). Closely spaced NPs are clearly seen, which we propose results from NP assemblies grown on the template molecules. These NP pairs exist alongside other NPs on the TEM grid grown in empty droplets.

To provide further evidence for the above claims, we have performed an interparticle-distance analysis for identified NP pairs, giving rise to mean interparticle distances of 0.96 and 1.09 nm for the hexanediol and octanediol cases, respectively (see Fig. 3d, e for the statistics). Density functional theory (DFT) calculations were then performed to reveal the global energy minimum structures of Au–hexanediol–Au and Au–octanediol–Au clusters by using the Gaussian 16 quantum calculation package[39]. The DFT-D3 calculations employed the B3LYP functional, with 6-311 + G(d,p) basis sets for C, O, and H, and a LANL2TZ(f) ECP basis set for Au atoms. Au atoms are found to bind preferentially to the O atoms (see Figs. 3f, g and S1), with the distance between two Au atoms at the O atoms being 1.071 and 1.165 nm for Au–hexanediol–Au and Au–octanediol–Au, respectively (see Fig. 3f, g). By performing counterpoise corrections, the calculations also reveal the corresponding Au-O binding energies of 18.7 and 18.9 kJ/mol, which are relatively weak interactions, for example, when compared to the hydrogen bonding in a water dimer[40]. Notably, the average interparticle distances are quite close to the calculated distances between two Au atoms in the binary clusters. The marginally shorter distances can be attributed to the attraction between the Au nanoparticles within the pairs at short distances and the relatively low rigidity of the hydrocarbon chains in diol molecules. These provide additional evidence that NP assemblies have been formed by the addition of molecular templates and that sub-nm precision for the interparticle distances has been achieved.

To further confirm the formation of NP assemblies by using diols as the templates, we have also performed a nearest-neighbour distance (NND) analysis, with and without template molecules, by using the ImageJ plugin[41]. In the NND analysis, a particle is selected as being at the origin and the probability for the existence of any neighbouring particles is presented as a function of distance. The NND plots with 1,6-hexanediol and 1,8-octanediol templates are shown in Figs. 3i and 3j, respectively. To generate these data, we analysed TEM images comprising more than 600 NPs in each case, and out of these, we found 105 closely spaced pairs for the hexanediol experiments and 124 pairs for octanediol. The addition of diols to helium droplets has clearly resulted in bi-modal distributions, with the sharp peaks at relatively short interparticle distances corresponding to the nanoparticle pairs formed by using diol molecules as the templates. The broad features in Fig. 3i, j are attributed to helium droplets that fail to pick up diol molecules, which are identical to those in Fig. 3h (without the addition of diol molecules). The NND analysis also provides a direct measure of the fraction of nanoparticles in the assemblies, i.e., with a percentage of 17.5% and 20.6% for 1,6-hexanediol and 1,8-octanediol, respectively. Consequently, the interparticle distance and NND analyses confirm that the structures of NP assemblies are determined by binding sites within the molecular templates.

It is noteworthy that the paired NPs often show a difference in sizes (see inset of Fig. 3b, c). We tentatively attribute this to a steering effect by Au atoms picked up early by the droplets, which can preferentially attract further Au atoms because of the collective polarizability of that Au cluster. In

other words, although the pick-up of Au atoms by the droplet is entirely stochastic, there is often an in-built bias in which the larger Au cluster acts as a preferred nucleation centre.

Since the shrinkage of a helium droplet by the pickup of a diol molecule is negligible relative to their initial size, the total Au content in helium droplets should be identical with and without the templates (see Fig. 4a) and this is clearly demonstrated by analysing the particle sizes. Notably, the diameters of NPs follow a Gaussian distribution (see Fig. 4b, c). For 1,6-hexanediol, the unpaired NPs have an average diameter of 4.2 nm, which is greater than those in NP pairs (an average diameter of 3.0 nm). For 1,8-octanediol, the average particle size is 6.4 nm for unpaired particles formed in empty droplets; while the average diameter of all the particles in the NP pairs is 4.3 nm. Nevertheless, as illustrated in Fig. 4d, e, the total Au content in empty droplets and those having molecular templates show identical size distribution within the margin of error, providing further evidence that the nanoparticle pairs identified are a consequence of the pre-addition of diol molecules to helium droplets. Note that we have employed slightly different oven temperatures for Au, i.e., 1220 and 1250 K for 1,6-hexanediol and 1,8-octanediol, respectively, resulting in different NP contents in Fig. 4d, e. A higher oven temperature leads to a higher partial pressure of Au vapour and, thus, larger Au NPs, which is clearly evident in the findings.

### Porphyrin-templated nanoparticle assemblies

The other template employed in this study is 5,10,15,20-tetra(4-pyridyl) porphyrin (H2TPyP). H2TPyP has four pyridyl N atoms located at the vertices of a square, which are expected to be the preferred binding sites for

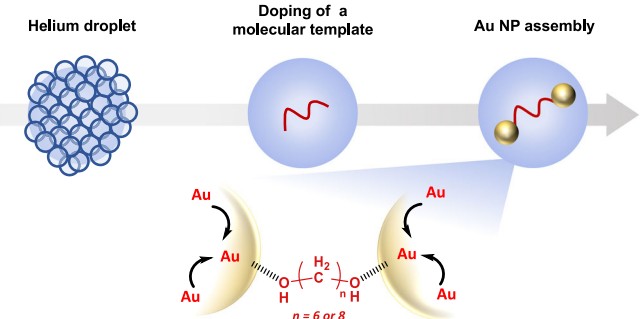

**Fig. 1 | Proposed mechanism for the formation of nanoparticle assemblies in helium droplets by using a molecular template.** As helium droplets travel in the vacuum through sample cells, they first acquire a molecular template (a diol molecule in this illustration) and then metal atoms to form nanoparticles on specific binding sites of the templates.

Au atoms. In this case, the template will allow the construction of more elaborate two-dimensional structures. In this experiment, a higher template doping rate was employed than in the diol experiments, with the result that most droplets contained more than one H2TPyP molecule. Technically, this increases the number of binding sites in helium droplets, allowing a greater number of particles in the assemblies. The TEM image and the statistics of the resulting structures are presented in Fig. 5. Interestingly, a few rhombus-shaped nanoparticle assemblies can be identified, which have a similar arrangement to that expected for the four pyridyl nitrogen atoms in H2TPyP (see Fig. 5a). Note that the N atoms do not form a perfect square in H2TPyP because the pyridyl rings are tilted away from a planar arrangement owing to steric strain. However, most NPs align into a chain structure, which we attribute to the presence of more than one H2TPyP molecule per helium droplet. The NND analysis is shown in Fig. 5b, giving an average interparticle distance of 1.37 nm.

DFT calculations on the H2TPyP-Au$_4$ cluster, with one Au atom attached to each pyridyl N atom, predict an average distance of 1.42 nm between neighbouring Au atoms and a binding energy of 45.02 kJ/mol between Au and the pyridyl N atom, as seen in Fig. 5c. Again, this demonstrates excellent agreement between the experimental interparticle distance and that calculated using DFT. Additionally, DFT calculations on the H2TPyP dimer have also been performed, which show a chain structure with double hydrogen bonds between two pairs of the pyridyl N atom and pyrrole H atom (see Fig. S2). Such a structure allows Au atoms to aggregate on the exposed pyridyl N atoms and form nanoparticle chains, as observed by TEM imaging.

Compared with diol-templated nanoparticle assemblies, the mean gold particle size is smaller (~2.48 nm) for H2TPyP-templated nanoparticle assemblies (see Fig. 5d), which can be attributed to the additional binding sites within the template molecule. The NP size distribution is also narrower compared to that of diol-templated assemblies, which can be explained by the larger spacing between the binding sites and, thus, the reduced steering effect from existing Au clusters to incoming Au atoms and the stronger interaction between Au atoms and the pyridyl N atom. It is noteworthy that, on average, the nanoparticle assemblies contain 4 Au NPs, as seen in Fig. 5e. Most Au NPs are found in assemblies, with an assembling efficiency of ~92%, which is consistent with the higher doping rate of H2TPyP used in this experiment.

### Conclusions

We have shown for the first time that it is feasible to grow metal nanoparticle arrays using molecular templates in which the particle position is dictated by weak van der Waals forces. This has been achieved through the sequential addition of a molecular template and Au atoms into helium nanodroplets,

**Fig. 2 | Schematic illustration of the apparatus used for the growth of Au nanoparticle assemblies.** **a** 5 μm pinhole nozzle; **b** skimmer; **c** sample cell for adding the molecular template; **d** molecular template container used to supply the molecular template by passage into the vacuum through a needle valve; **e**, ceramic oven for evaporating gold, which is resistively heated by a tantalum wire; **f** deposition station for manipulation and removal of the TEM grid; and **g** quadrupole mass spectrometer for beam diagnostics.

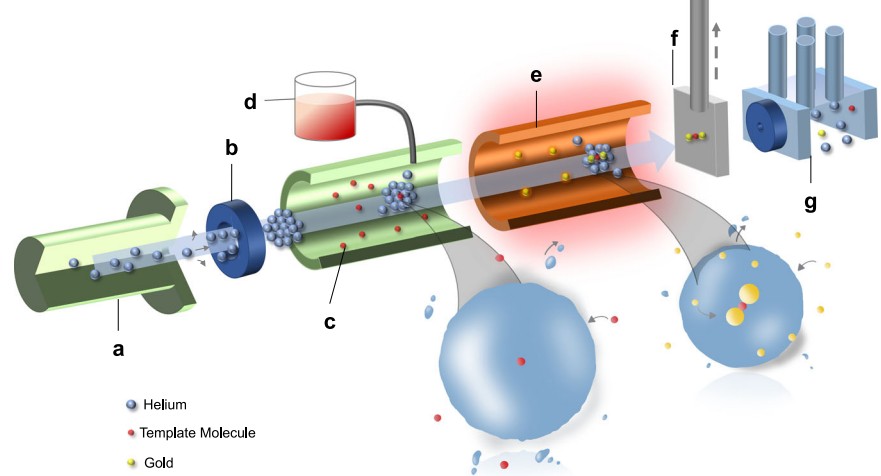

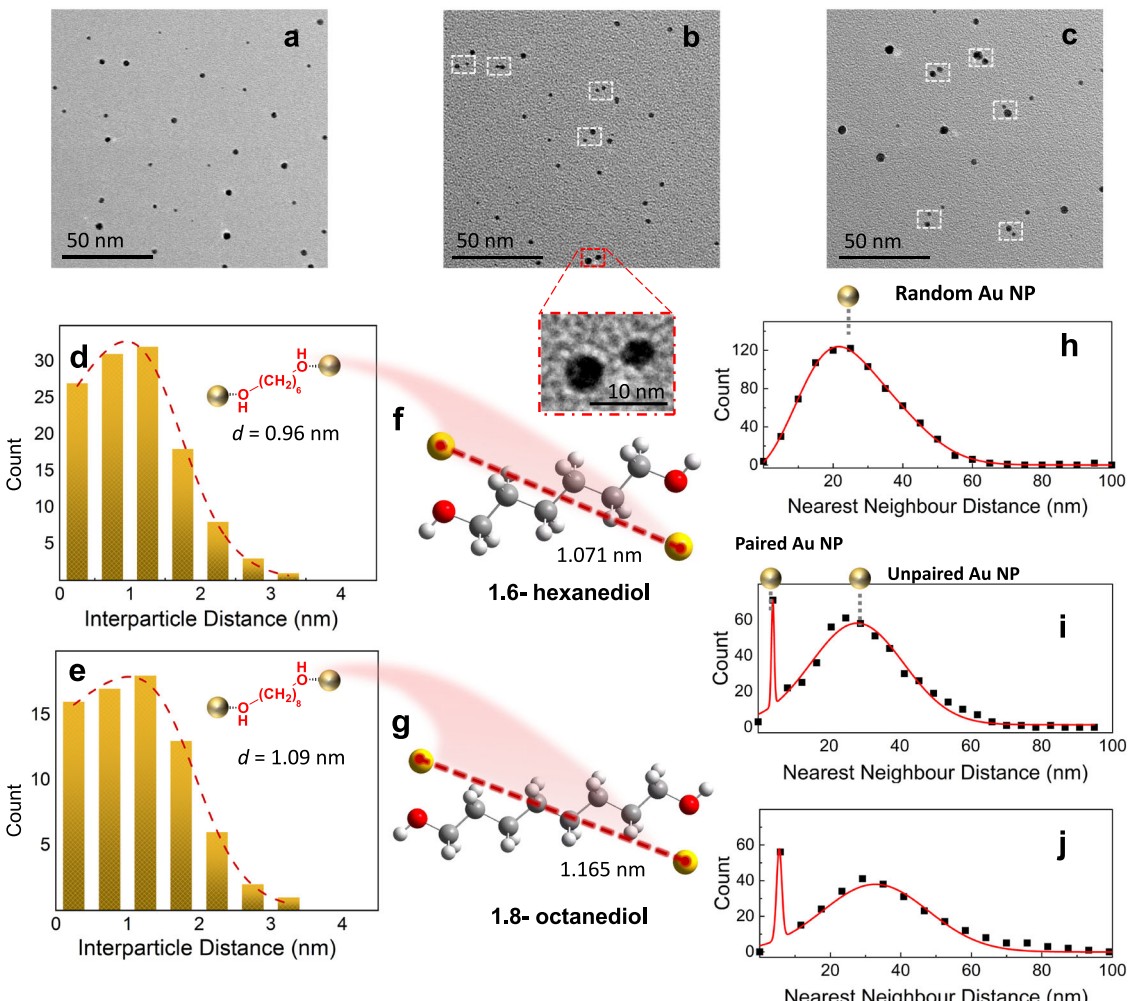

**Fig. 3 | TEM imaging derived from deposition of Au-doped helium droplets, with and without the addition of molecular templates. a** typical TEM image of Au nanoparticles formed in HeDs without the addition of diols. **b** 1,6-hexanediol as the template. **c** 1,8-octanediol as the template. **d, e** interparticle distance analysis for 1,6-hexanediol and 1,8-octanediol-templated particle pairs. The mean interparticle distances (*d*) are 0.96 nm and 1.09 nm for hexanediol- and octanediol-templated particle pairs, respectively. **f, g** global energy minimum structures of Au–hexanediol–Au and Au–octanediol–Au clusters obtained by DFT calculations. **h** Nearest-neighbour distance (NND) plot of Au nanoparticles formed in empty HeDs. **i, j** NND plots with 1,6-hexanediol and 1,8-octanediol as templates, respectively. In images **b** and **c**, the dashed lines highlight possible nanoparticle pairs formed by adding a molecular template to the HeDs. The curves (solid lines) in images **d, e** and **h–j** were obtained by fitting to a Gaussian distribution.

followed by collision with a surface. The low temperature of the growth medium (superfluid helium) and the high mobility of atoms and molecules within this environment are believed to be central to these successful observations. We show that molecular templates can offer precise control over the interparticle distances and structures of the nanoparticle assemblies. These initial findings need further exploration with a far wider variety of templates to show the general versatility, flexibility and applicability of this approach.

Although limitations exist on the volatility of molecular templates and the maximum size of particles, the superfluid helium templating approach has implications for applications in areas such as molecular sensors and quantum information transport, which rely on the optical response of and coupling between plasmonic nanoparticles with the key being the control over the particle sizes and interparticle distances. Potentially this technique can be used to enable precision assembly of a wide range of nanomaterials, including plasmonic, semiconductor, and hybrid nanoparticles, with tailored optical, electronic, magnetic, and catalytic properties. For example, plasmonic nanoparticles with controlled spacing would allow tuning of optical resonances for enhanced spectroscopy and sensing. The versatility of the technique could also lead

it to be used to create novel metastable nanostructures, hence yielding new opportunities to engineer nanomaterials and devices with capabilities beyond conventional self-assembly methods. In addition, our approach meets the constant need for miniaturization technology, moving away from conventional pick-and-place methods and towards assembly methods based on natural self-organization principles. In this regard, this study illustrates a strategy for control of the size, structure and interparticle distances in the nanoparticle assemblies at a sub-nm precision, opening up a new avenue for the rational design of plasmonic and photonic devices.

## Methods

### Density functional theory (DFT) calculations

To predict the structures of nanoparticle assemblies formed by the sequential addition of selected molecular templates, namely, 1,6-hexanediol, 1,8-octanediol or H2TPyP, and Au atoms to helium droplets and hence validate our new approach, we have performed DFT calculations at the B3LYP-D3 level of theory to obtain the global energy minimum structures of 1,6-hexanediol, 1,8-octanediol, Au–hexanediol–Au, Au–octanediol–Au, H2TPyP, H2TPyP-Au$_4$ and the H2TPyP dimer. These are achieved by using

**Fig. 4 | Nanoparticle sizes and Au content in empty HeDs and those with molecular templates.** **a** Schematic presentation of the resulting nanostructures with and without a template, both having the same Au content. **b**, **c** Particle size distributions using 1,6-hexanediol and 1,8-octanediol as the templates, respectively. The curves were fitted to a Gaussian distribution; **d**, **e** Total number of Au atoms of unpaired nanoparticles (yellow circles) and nanoparticle pairs (red triangles) in the presence of 1,6-hexanediol and 1,8-octanediol, respectively. The curves were obtained by fitting them to a log-normal distribution.

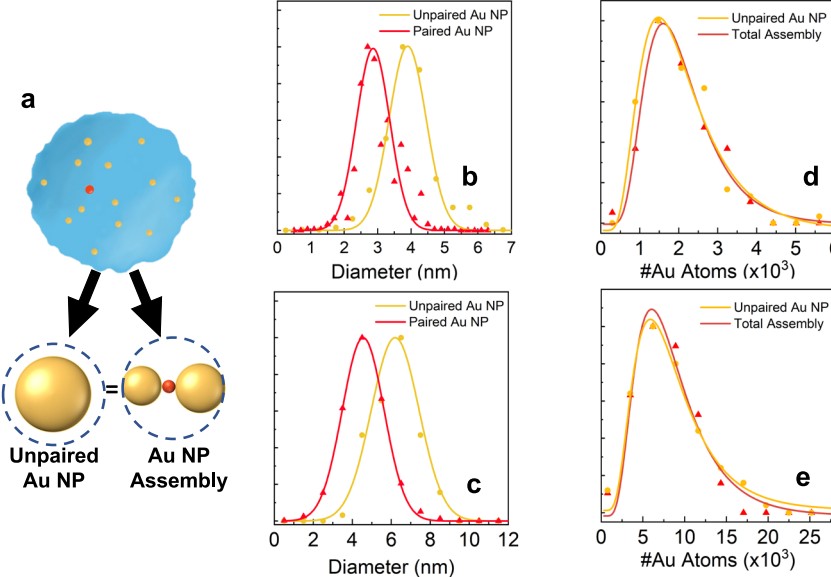

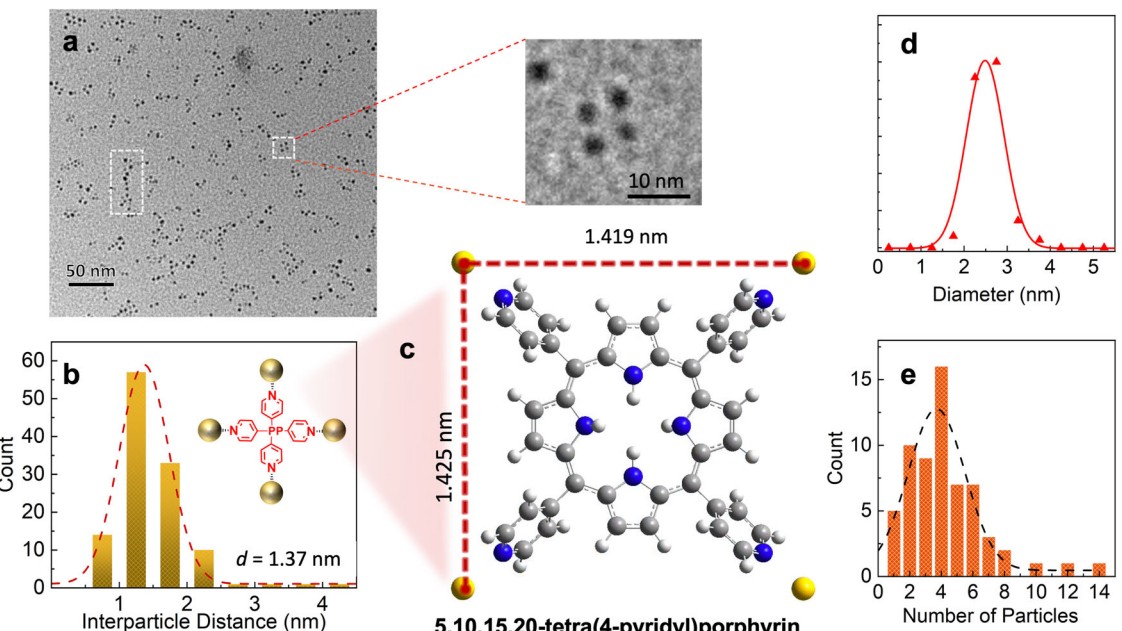

**Fig. 5 | Growth of nanoparticle assemblies using H2TPyP as the template. a** TEM image. **b** Interparticle distance analysis, with a mean interparticle distance of 1.37 nm. **c** The structure of H2TPyP-Au$_4$ cluster obtained by DFT calculations. **d** Particle size distribution and **e** number of nanoparticles per assembly. A Gaussian distribution was applied for the curve fittings in images **b**, **d** and **e**.

the Gaussian 16 quantum chemical computation package[39]. The basis sets used are the 6-311 + + G(d,p) basis sets for O, C, N and H, and the LANL2TZ(f) ECP basis set for Au. The optimized geometries of these clusters are presented in Figs. S1 and S2 and Supplementary Data 1. To evaluate the binding energies between Au atoms and the molecular templates, counterpoise corrections have been performed to account for basis set superposition errors.

## Data availability
Any relevant data are available from the authors upon reasonable request.

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

## Acknowledgements

S.Y. wishes to thank the UK EPSRC (grant codes: EP/J021342/1; EP/V027255/1) and the Leverhulme Trust (Grant codes: RPG-2016-272 and RPG-2020-152) for funding this work.

## Author contributions

G.T. analysed the data and prepared the draft of the manuscript. A.M.A.A.H., C.F., D.S., B.S. and H.L. contributed to the experiments and collected experimental data. A.M.E. assisted with the manuscript preparation. S.Y. designed and conceptualized the research.

## Competing interests

The authors declare no competing interests.
