## [Peer Review File · Communications Chemistry]

Reviewers' comments:

Reviewer #1 (Remarks to the Author):

Over the last decade, coagulation of impurities picked up by superfluid helium nanodroplets has been found to offer a unique, interesting, and flexible technique for nanoparticle synthesis and surface deposition. Particles of variable sizes, core-shell particles, and nanowires grown along vortex lines have been produced.

In this work, the authors demonstrate a novel way to grow and organize nanoparticles within nanodroplets: capturing them at the attractive tips of molecular templates. They exhibit it by first embedding within nanodroplets either two types of linear chain molecules (diols) or a square-like porphyrin. Imaging clearly shows the formation of metal nanoparticle pairs in the former case and quartets of nanoparticles in the latter case.

The paper is clearly and logically written, and provides a well-presented sequence of measurements adding up to a convincing confirmation that the proposed mechanism is really working. The concept is elegant and offers a lot of potential flexibility and tunability. It will be of interest to a range of readers and will certainly stimulate a lot of follow-up work.

As a result, the paper is well suited for publication in its present form. There are some minor suggestions to the authors to elaborate on a few points, at their discretion. :

1. The paragraph comparing particle size plots in Fig 3 reads "For 1,8-octanediol, the average particle size is 6.4 nm for unpaired particles formed in empty droplets; while those in the NP pairs have an average diameter of 4.3 nm." At first reading, it is not completely clear whether the 4.3 nm refers to the diameter of each particle in the pair, or to some kind of combined diameter of both. Almost certainly it's the former, but a clearer sentence would make this easier to catch at first reading.

2. Near the end of the paper and in the supplemental material the authors address the likelihood that H2PyP form dimers within the nanodroplet and these dimers then serve as nanoparticle growth templates. Might the diols form dimers as well?

3. If space allows, it might be helpful to make panel A of fig. 4 larger. It is a crowded image and it is difficult to make out the highlighted features in white frames and to compare them with the surroundings.

Reviewer #2 (Remarks to the Author):

In this manuscript Yang et al. reported on tailoring of nanoparticle assemblies inside helium nanodroplets, which were subsequently deposited on a surface and characterized/visualized using TEM. The approach of using weak van der Waals interactions between gold and diols or porphyrins for templating in such an inert medium is promising and this study is in my opinion a good proof-of-concept. The authors meticulously rationalized the absence of droplet size change upon doping in the experimental set-up used, described measures taken to prevent unwanted aggregation and could compare the nanoparticles formed both from “empty” droplets and those containing a template. These aspects are important since they ensure that the observed sizes and arrangements of nanoparticles originating from doped helium nanodroplets are a direct consequence of the templating effect of the introduced organic molecules. The results are presented concisely and the scientific investigation was solidly done. The manuscript is written clearly, is easy to follow and the style is of good quality. However, in continuation are some aspects that should be addressed before publishing:

1. The authors note that nanoparticle pairs formed using diol templates often differ in sizes, i.e., two pair members are of different size. An explanation is given that larger gold clusters act as a preferred nucleation center, hence the final mismatch. From the provided TEM images it can indeed be seen that such pairs demonstrated a significant variety in size ratios of pair members. How do the authors comment this from a perspective of desired precise control over the structures of such nanoparticle assemblies? Could there be an experimental way to introduce more control during the growth/assembly process (e.g., concentration, temperature variations during different helium nanodroplet doping stages) that would in the end result in more uniform pair structures?

2. DFT calculations were performed to confirm the proposed interactions between the O or N atoms and gold. The used level of theory included the use of D3 correction important for systems relying on weak van der Waals interactions to hold the supramolecular structure in place. However, the authors considered only one gold atom per binding site, presumably due to the high cost of computational time and resources for more elaborate gold structures. However, could it be possible to perform at least one calculation with a smaller sized gold cluster complex (e.g., 1,6-hexanediol with two icosahedral Au₁₃ clusters or smaller) to validate the conclusions of the presented one-atom approach?

3. In the supporting information the authors provided only figures of the calculated structures. It would be useful to also include the xyz coordinates, either as a table or as separate files.

4. The authors report on a DFT-calculated Au–O interaction energy of 18.7 kJ/mol and 18.9 kJ/mol for the respective diols. Could they briefly describe the process of how they obtained these values in order to be more informative to the readers? Additionally, the corresponding Au–N interaction energy for porphyrin derivative is not mentioned. Could the authors report this value as well and then comment on the O vs. N binding strengths for these two classes of template molecules?

It is my opinion that this manuscript could be of interest to the readership of Communications Chemistry and could point towards a new way of using weak non-covalent interactions for supramolecular nanoparticles assembly. I therefore recommend it for publication after the authors address these points.

Reviewer #3 (Remarks to the Author):

The authors have presented a novel method for the self-assembly of Au nanoparticles (NPs) using diols and porphyrin as molecular templates in a superfluid helium droplet environment. The TEM results clearly show that the Au NPs are deposited on the grid with a statistically significant pattern indicating the NPs are grown at favorable binding sites. The experiment results are also consistent with the DFT calculation, though, in both diol and porphyrin cases, the measured pair distances of Au NPs are slightly smaller than the calculated ones. This work demonstrates that superfluid helium droplets can serve as a unique environment to grow NPs in a controlled manner which could lead to precise control of interparticle distances in growing NPs. This is an exciting work, and it could greatly impact the nanoparticle synthesis and superfluid helium research field.

I have a few questions/concerns regarding the experimental data and analysis:

1. when using 1,6-hexanediol and 1,8-octanediol as templates, why there is a significant difference in the average Au NPs diameters? 6.4 nm vs 4.2 nm in the unpaired particles in empty droplets and 4.3 nm vs 3.0 nm in the paired particles. Is this difference purely caused by the different oven temperatures (1220 K vs 1250 K)?

2. Based on the most probable Au atom numbers in the study of 1,8-octanediol template, after doping 6000 – 7000 Au atoms, the superfluid helium's size should be significantly reduced if it can survive. My understanding is that the 6.4 nm average NP size is determined by the initial droplet size. Was there any experiment done at a different source temperature?

3. It will be better to provide TEM grid type and instrumental information, at least in the supplemental documents. Could it be possible for the complexes to land on the TEM grid at a different angle than lying

flat? In the porphyrin template study, there is a pattern on the TEM image that three Au NPs with fixed interparticle distances aligned in a linear format. Is there a good model to explain this pattern? While the dimer structure can explain the chain shape formation, it should lead to a double chain instead of a single linear chain.

Dear Editors,

Thank you so much for sending us the feedback. We are pleased to address the minor concerns of the reviewers, as detailed below, and any changes made in the main manuscript are highlighted below in red.

Reviewer #1:

1. The paragraph comparing particle size plots in Fig 3 reads “For 1,8-octanediol, the average particle size is 6.4 nm for unpaired particles formed in empty droplets; while those in the NP pairs have an average diameter of 4.3 nm.” At first reading, it is not completely clear whether the 4.3 nm refers to the diameter of each particle in the pair, or to some kind of combined diameter of both. Almost certainly it’s the former, but a clearer sentence would make this easier to catch at first reading.

A: We have altered the relevant sentence to say “**while the average diameter of all the particles in the NP pairs is 4.3 nm**”.

2. Near the end of the paper and in the supplemental material the authors address the likelihood that H₂PyP form dimers within the nanodroplet and these dimers then serve a nanoparticle growth templates. Might the diols form dimers as well?

A: Yes, diols might form dimers but we have kept the diol doping rates so low that dimer formation is improbable (estimated to be less than 5% of the monomers).

3. If space allows, it might be helpful to make panel A of fig. 4 larger. It is a crowded image and it is difficult to make out the highlighted features in white frames and to compare them with the surroundings.

A: We have now added an inset to show a magnified image of a nanoparticle assembly containing 4 Ag nanoparticles in Fig. 4.

Reviewer #2:

1. The authors note that nanoparticle pairs formed using diol templates often differ in sizes, i.e., two pair members are of different size. An explanation is given that larger gold clusters act as a preferred nucleation center, hence the final mismatch. From the provided TEM images it can indeed be seen that such pairs demonstrated a significant variety in size ratios of pair members. How do the authors comment this from a perspective of desired precise control over the structures of such nanoparticle assemblies? Could there be an experimental way to introduce more control during the growth/assembly process (e.g., concentration, temperature variations during different helium nanodroplet doping stages) that would in the end result in more uniform pair structures?

A: this is a really good comment. This is best answered by reference to the TPyP templated nanoparticle assemblies. In this case we have achieved a significantly narrower size distribution (~1 nm for FWHM). From the TPyP experiment, we speculate that slightly longer distance between the binding sites, e.g., from ~1 nm to 1.5 nm, is sufficient to produce near monodisperse nanoparticles.

2. DFT calculations were performed to confirm the proposed interactions between the O or N atoms and gold. The used level of theory included the use of D3 correction important for systems relying on weak van der Waals interactions to hold the supramolecular structure in place. However, the authors considered only one gold atom per binding site, presumably due to the high cost of computational time and resources for more elaborate gold structures. However, could it be possible to perform at least one calculation with a smaller sized gold cluster complex (e.g., 1,6-hexanediol with two icosahedral Au₁₃ clusters or smaller) to validate the conclusions of the presented one-atom approach?

A: The reviewer is correct that we considered only one gold atom in the calculations because of the high cost of computational time and resources needed for larger gold clusters. Owing to the lack of computational time we are currently unable to consider larger Au clusters.

3. In the supporting information the authors provided only figures of the calculated structures. It would be useful to also include the xyz coordinates, either as a table or as separate files.

A: We have now added the XYZ coordinates for the obtained optimized structures of the molecular clusters in the SI.

4. The authors report on a DFT-calculated Au–O interaction energy of 18.7 kJ/mol and 18.9 kJ/mol for the respective diols. Could they briefly describe the process of how they obtained these values in order to be more informative to the readers? Additionally, the corresponding Au–N interaction energy for porphyrin derivative is not mentioned. Could the authors report this value as well and then comment on the O vs. N binding strengths for these two classes of template molecules?

A: We used the standard counterpoise-corrected procedure to calculate binding energy in Gaussian. This has now been added to the manuscript by adding a half sentence in the final paragraph of Page 6: “**By performing counterpoise corrections ...**”

Counterpoise corrections give an Au-TPyP binding energy of 45.0 kJ/mol, which is significantly stronger than that of Au-diol binding energy. Accordingly, the interatomic distances of Au-O and Au-N bonds are 2.685 and 2.315 Å, respectively, with the former

being significantly longer. The stronger Au-N interaction is also indicative for stronger short-range interactions between Au atom and the pyridyl group, which can reduce that steering effect caused by Au atoms – an additional factor contributing to the narrower size distribution of Au nanoparticles within the assembly. We have added this information to the main text (second last paragraph in Page 8).

Reviewer #3:

1. when using 1,6-hexanediol and 1,8-octanediol as templates, why there is a significant difference in the average Au NPs diameters? 6.4 nm vs 4.2 nm in the unpaired particles in empty droplets and 4.3 nm vs 3.0 nm in the paired particles. Is this difference purely caused by the different oven temperatures (1220 K vs 1250 K)?

A: Yes, the difference is purely caused by the temperature difference of the Au oven in the two different experiments.

2. Based on the most probable Au atom numbers in the study of 1,8-octanediol template, after doping 6000 – 7000 Au atoms, the superfluid helium's size should be significantly reduced if it can survive. My understanding is that the 6.4 nm average NP size is determined by the initial droplet size. Was there any experiment done at a different source temperature?

A: The reviewer is correct regarding the size reduction while adding more metal atoms. However, the size of the particles is determined not only by the initial droplet size, but also the Au vapor pressure (which in turn is determined by the oven temperature).

From our past experience the particle size increases monotonically with the oven temperature until the doping reaches a limit at which all of the helium atoms within the droplets are vaporized.

3. It will be better to provide TEM grid type and instrumental information, at least in the supplemental documents. Could it be possible for the complexes to land on the TEM grid at a different angle than lying flat? In the porphyrin template study, there is a pattern on the TEM image that three Au NPs with fixed interparticle distances aligned in a linear format. Is there a good model to explain this pattern? While the dimer structure can explain the chain shape formation, it should lead to a double chain instead of a single linear chain.

A: We use standard lacey carbon TEM grids for this work with a thickness of ~10 nm and a diameter of 3 mm. This information has been added to the main text at the first paragraph of Page 4.

The detailed landing dynamics is unclear. The reviewer is right that in principle, some of the particle pairs may land at angle rather than flat (parallel to the surface for the assemblies). Indeed, similar phenomenon has been observed and reported by different groups, e.g., the formation of one-dimensional nanostructures (see *Phys. Rev. B* **90**, 155442 (2014) and *Nano Lett.* **14**, 2902-2906 (2014), for example).

The reviewer also makes a very good point regarding the linear alignment of particles when using TPyP as the template, which we speculate is due to the formation of TPyP cluster chains and have performed a calculation for TPyP dimer accordingly. However, this cannot explain why a single chain is formed rather than double chain, as suggested by the reviewer. A further investigation, including both experimental and theoretical modelling work, is necessary in order to provide a definitive answer.

REVIEWERS' COMMENTS:

Reviewer #2 (Remarks to the Author):

In my opinion the authors sufficiently addressed the raised concerns and I therefore recommend the paper for publication.

Reviewer #3 (Remarks to the Author):

The authors have address all my points and I have no further questions.